# Computer Modelling of Energy Structure of Yb^3+^ and Lu^3+^ Doped LaF_3_ Crystals

**DOI:** 10.3390/ma15227937

**Published:** 2022-11-10

**Authors:** Yaroslav Chornodolskyy, Vladyslav Karnaushenko, Jaroslaw Selech, Vitaliy Vistovskyy, Taras Demkiv, Krzysztof Przystupa, Stepan Syrotyuk, Anatolii Voloshinovskii

**Affiliations:** 1Faculty of Physics, Ivan Franko National University of Lviv, 8 Kyryla i Mefodiya, 79005 Lviv, Ukraine; 2Institute of Machines and Motor Vehicles, Poznan University of Technology, Piotrowo 3, 60-965 Poznan, Poland; 3Department of Automation, Lublin University of Technology, Nadbystrzycka Str. 36, 20-618 Lublin, Poland; 4Department of Semiconductor Electronics, Institute of Telecommunications, Radioelectronics and Electronic Engineering, Lviv Polytechnic National University, 12 S. Bandera Str., 79013 Lviv, Ukraine

**Keywords:** lanthanide ions, band structure, projector augmented wave method, energy band gap

## Abstract

The energy band structure, as well as partial and total densities of states have been calculated for LaF_3_:Yb and LaF_3_:Lu crystals within density functional theory using the projector augmented wave method and Hubbard corrections (DFT + U). The influence of geometric optimization on the results of energy band calculations of LaF_3_:Ln crystals (Ln = Yb, Lu) was analysed and the absence of relaxation procedure is confirmed to negatively influence the energy position of states, and the variability between obtained results of different optimization algorithms are within the calculation accuracy. The top of the valence band of LaF_3_ is confirmed to be formed by the 2pF^-^-states and the bottom of the conduction band is formed by the 5d-states of La^3+^. The positions of the 4f-states and 5d-states of activator ions in LaF_3_ were studied. It is shown that the 4f-states of Yb^3+^ are slightly above the top of the valence band and the 4f-states of Lu^3+^ to be 3.5 eV below the top of the valence band. The energy levels of the 5d states of the impurities are energetically close to the bottom of the LaF_3_ conduction band. The calculated band gap of 9.6 eV for LaF_3_ is in a good agreement with the experimental result and is not affected by impurity ions.

## 1. Introduction

There is significant interest to study inorganic scintillators activated by the lanthanide elements because of the wide range of their usage as detectors in high energy physics, medicine and radiation safety [1]. The main luminescent properties of lanthanide ions (Ln^3+^) are determined by two types of transitions: intraconfigurational 4f → 4f and interconfigurational 4f → 5d, which can be characterized by the density of states and the energy of levels between which the transition occurs. Although LaF_3_:Yb and LaF_3_:Lu crystals have not found wide practical use, knowledge of their energy properties allows supplementing the popular empirical model of Dorenbos [2] with theoretical information that is required to better understand the scintillation processes in this type of crystals.

Nowadays, there is very limited information about the energy band calculation, and luminescence processes involving the Lu and Yb ions. This is due to the fact that the expected energy of 5d → 4f radiative transition is in the VUV range (of the order of 8–10 eV [3]) and requires matrices with a large band gap for observations. The most suitable matrices for studying luminescent processes involving Lu^3+^ and Yb^3+^ ions are fluorides of rare earth elements.

The most similar crystals to the studied LaF_3_-Lu with respect to energy structure are the experimentally studied LuF_3_:Ce and LiLuF_3_:Ce crystals. The energy properties of these crystals were studied in [3], where the 4f^13^5d→4f^14^ transition of the Lu^3+^ ion was found to be in the vacuum ultraviolet region, and the 4f-states are located near the top of valence band, increasing the efficiency of thermalized holes capture. The LuF_3_-Ce is also characterized by the absence of the zero-phonon line, which indicates a strong electron–phonon interaction [4]. The 4f-states of Lu^3+^, despite their energy proximity to the valence band of the LuF_3_ crystal, do not participate in the formation of chemical bonds with ligands [5]. The direct studies of the luminescent transitions in Lu^3+^ ions were performed in the matrices of LiLuF_4_, LuF_3_, LiYF_4_:Lu^3+^ [6] and LiCaAlF_6_ [7]. VUV emitting 5d-4f spin-forbidden transitions in the Lu^3+^ ion from high-spin 5d states in wide-band LuF_3_ and in LiYF_4_:Lu^3+^ (1.0 %) are observed at low temperatures. The processes of quenching the 5d-4f luminescence of Lu^3+^ ions occur due to temperature-stimulated transitions of electrons from the 5d states of the Lu^3+^ ion with low spin to the conduction band. The latter indicates the close location of the 5d states of the Lu^3+^ ion to the bottom of the conduction band of the crystals.

Experimental studies of the LaF_3_:Yb crystal were carried out for a long time, because the Yb^3+^ ion found wide practical usage as a sensitizer in the case of upconversion luminescence [8]. The Yb^3+^ ions in different matrices were also studied as an example of luminescence in the case of charge transfer from ligand to Yb^3+^ ion [9]. The positions of 4f and 5d levels in different matrices were analysed in papers where 4f-levels Yb^3+^ were shown to be near the top of the valence band and 5d-levels in the region of exciton levels of the LaF_3_ matrix [2].

Due to the limited number of theoretical calculations of the 4f and 5d level positions in the LaF_3_ matrix and the need for choosing the optimum theoretical approach for energy level calculations of the lanthanide ions, we propose to use the method that has proven its efficiency for calculating the energy levels of cerium ions in the LaF_3_ and CeX_3_ matrices [10,11,12]. Verification of the calculations will be carried out by comparing them with the determined positions of the levels within the empirical model of Dorenbos [2].

## 2. Materials and Methods

The calculation of the electronic properties of the studied crystals, such as the partial and total densities of states, as well as the band-energy structure was performed within the density functional theory (DFT) [13] using the projector augmented wave method (PAW) [14] and Hubbard U parameter (DFT + U) [15]. The Hubbard screened energy U is used to describe the strongly correlated 5d electrons of La and 4f electrons of Yb and Lu. Its use leads to more correct positioning of the energy levels of the mentioned electrons. This approach was repeatedly tested to study the energy properties of the LaF_3_:Ln^3+^ crystals [16], and, in comparison to the local density approximation (LDA), provides more accurate 4f-level positions and band gap values.

The LaF_3_ crystal unit cell consisted of 24 atoms. The optimization of the structural parameters of the crystal was carried out in two stages. At the first stage, the minimum of the total energy was found and the new values of the crystal lattice parameters were recorded, as given in Table 1. At the second stage, the optimized reduced parameters of the coordinates of the atoms in the unit cell were found.

After the introduction of Yb and Lu impurities, the crystal structures were optimized again. After that, the electronic properties of the LaF_3_:Yb and LaF_3_:Lu crystals were calculated.

Computations of the computer models of the LaF_3_:Yb and LaF_3_:Lu crystals used LaF_3_ crystallographic data from the online resource “Materials Project” [17] and are provided in Table 1. Due to the limited computer resources for calculation of impurity energy levels, we chose a model for which one of the lanthanide ions of the matrix unit cell was replaced by the ytterbium or lutetium ion, respectively. To consider the difference between the ionic radius of lanthanum and ytterbium/lutetium, the procedure of geometry relaxation was performed. New ions’ positions were calculated with the same lattice parameters as in the original matrix.

Two approaches were chosen to study the influence of the geometric relaxation methods on the structure of newly formed lattices. The first approach is based on modifying the positions of atoms, keeping the volume and shape of the crystal lattice unchanged. Structural optimization was performed by the BFGS algorithm [18], which allows finding of the equilibrium position by minimizing the maximum value of the forces acting along the base vectors of the crystal cell. An alternative way of relaxation was performed by changing the lattice volume without any changes to atom positions. It is based on the modified BFGS algorithm [19], which considers not only the value of total energy, as in the original version of the algorithm, but also its gradients.

The abovementioned methods were implemented in open-source Abinit software. Hubbard corrections for PAW datasets of lanthanide elements were taken from paper [20].

The k-points basis was formed on the Monkhorst-Pack grid of 8 × 8 × 8. The plane wave basis was created with the cutting energy values of 817 eV and 3266 eV within and outside of the PAW region, respectively. The Hubbard corrections were 3 eV for Yb and 5.5 eV for Lu.

## 3. Results

### 3.1. The Influence of Crystal Cell Relaxation on Its Energy Properties

This section presents the study results of the effect of unit cell structure relaxation of the LaF_3_:Lu crystal on the energy properties after replacement of lanthanum ion in the original LaF_3_ matrix by the lutetium ion. Plots in Figure 1 show changes in the partial density of electronic states after applying various cell optimization methods, namely: (a) the effect of changes in ion positions when the lattice volume remains constant; (b) the effect of changing the volume of the lattice without changing the positions of ions; and (c) without cell relaxation. The results obtained by both optimization methods show almost the same values for the position of the 4f and 5d states of the Lu^3+^ ion and differ only in intensity (proportionally to all states). They are given in Figure 1a,b. On the other hand, the partial density of 4f-states, calculated using the unrelaxed crystal cell model, is 1 eV lower (Figure 1c) than the energy of 4f states in comparison with first two results.

The positions of 4f-states in relaxation models are closer to the experimental results, where 4f Lu^3+^ states are 1 eV below the top of the valence band [2], which makes the results obtained after geometric optimization by any of the algorithms more accurate than those obtained without relaxation of the structure. Such conclusions are also valid for the case of 4f-state location in LaF_3_:Yb.

The location of the 5d Lu^3+^ levels, relative to the conduction band, derived within the relaxed model show better comparison with the experiment [2].

### 3.2. Energy Properties of LaF3:Lu and LaF3:Yb

The plots in Figure 2 show the partial density of states of LaF_3_:Lu and LaF_3_:Yb crystals, respectively. In both crystals, the top of the valence band is formed by the 2p states of fluorine and the bottom of the conduction band is formed by the 5d states of lanthanum. The 5d states of impurity ions are located below the 5d states of lanthanum. The states of impurities are the most intense in these plots.

There is only one noticeable peak of 4f Lu^3+^ for LaF_3_:Lu with the energy of 3.5 eV in Figure 2a, which is 2.5 eV less than position of the states obtained from experimental data [2]. The 4f-states are most intensive. The width of the valence band is 3 eV and the lowest 5d Lu^3+^ states are placed below the conduction band formed by the 5d La^3+^ states.

In the case of LaF_3_:Yb (see Figure 2b), the position of 4f Yb^3+^ states with the energy of 0.3 eV higher than top of valence band converges with the experimental data with high accuracy [2]. The 5d Yb^3+^ states are partially localized near the bottom of the conduction band in an energy interval of 1.2 eV wide. The width of the valence band is 3 eV as in the case of LaF_3_:Lu.

The electronic band structures of the LaF_3_:Yb and LaF_3_:Lu crystals are shown in Figure 3 and Figure 4, respectively. The calculated band gap in both the cases is 9.6 eV, which agrees well with the experimental data [2] and with the calculation results obtained for LaF_3_, LaF_3_:Pm and LaF_3_:Sm [16,21,22]. If we compare the band structures of the studied crystals, we can see two key differences. The first is the presence of 4f state above the top of the valence band in LaF_3_:Yb and below the valence band in the case of LaF_3_:Lu. The second difference is the presence of a gap between the 5d Yb states and the bottom of the conduction band formed by the 5d La states in the proximity of Γ-point near 9.6 eV in the case of LaF_3_:Yb and small in the case of LaF_3_:Lu. It should be noted that small dispersion of energy 5d-bands located below the bottom of the conduction band indicates the local nature of the levels of impurity 5d-states. The revealed local character of lanthanide impurity states in the LaF_3_ matrix is characteristic of the energy states of the bottom of the conduction band of lanthanide halide crystals. Due to such local states in CeX_3_ (X = F, Cl, Br, I) crystals, the conditions for local states such as Frenkel excitons are formed [10,11].

The previous results were used to construct a comparative scheme between calculated energy levels of impurities in the LaF_3_:Yb and the LaF_3_:Lu crystals and experimental data (Figure 5). For comparison, a fragment (Figure 5a) of the scheme of lanthanide electronic levels in the LaF_3_ matrix was used [2]. The general tendency of the energy level locations is well reproduced by calculated results, namely the 5d-levels are located below the conduction band of the matrix, 4f-levels of Yb^3+^ are above the valence band, and 4f-levels of Lu^3+^ are located in the depth of the valence band.

## 4. Conclusions

The influence of geometric optimization on the calculation of electronic energy properties of LaF_3_:Ln crystals is analysed and it is confirmed that the lack of optimization negatively affects the position of the energy 4f levels of activator ions. The optimization of lattice by both the cell volume and the ion position conservation gives almost identical results.

The energy gap width in LaF_3_-Yb and LaF_3_-Lu crystals is determined by the distance between the 2p states of fluorine, which form the valence band, and the 5p states of lanthanum, which form the conduction band. It is 9.6 eV for both crystals. These values are in good agreement with the experimental data of 11.2 eV [2]. It was revealed that 4f levels of the Yb^3+^ ion in the studied matrix occupy the energy position near the top of the valence band and the 4f-states of Lu^3+^ ion are placed 3.5 eV below.

The positions of the 5d levels in LaF_3_:Yb and the LaF_3_:Lu crystals were shown to be energetically close and placed bellow the conduction band. The small energy gap between 2p F and 4f Yb states in the proximity of the Γ-point in case of the LaF_3_:Yb crystal was revealed.

The results are used to construct the comparative scheme between calculated energy levels of impurities in the LaF_3_:Yb and the LaF_3_:Lu crystals and the experimental data. The calculated positions of the 4f and 5d levels of the Yb^3+^ and Lu^3+^ ions in general qualitatively reproduce the experimental results.

## Figures and Tables

**Figure 1 materials-15-07937-f001:**
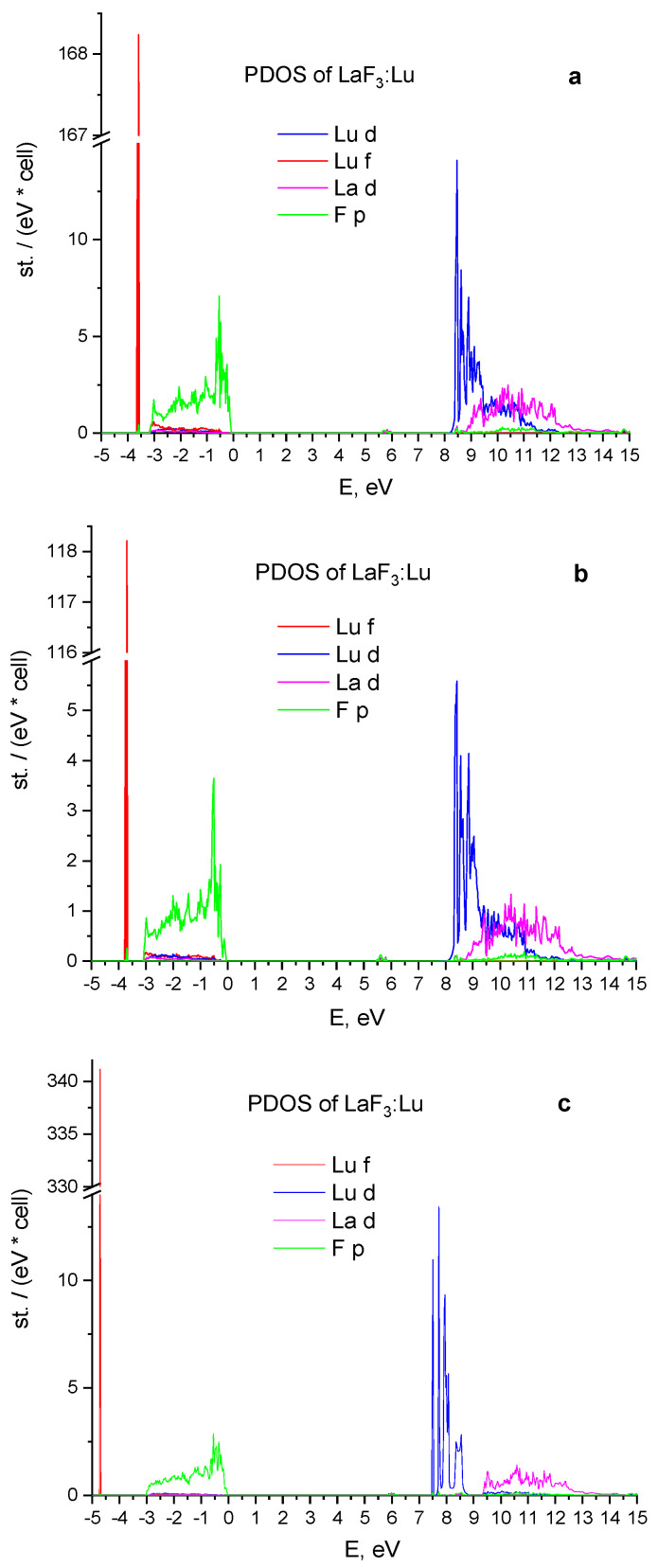
The influence of crystal structure optimization on the partial density of states (PDOS) after the substitution of the La ion by the Lu ion in the LaF_3_ host: (**a**) The change of the ion positions with fixed lattice volume; (**b**) the change of the lattice volume without changing of the ions position; (**c**) without relaxation.

**Figure 2 materials-15-07937-f002:**
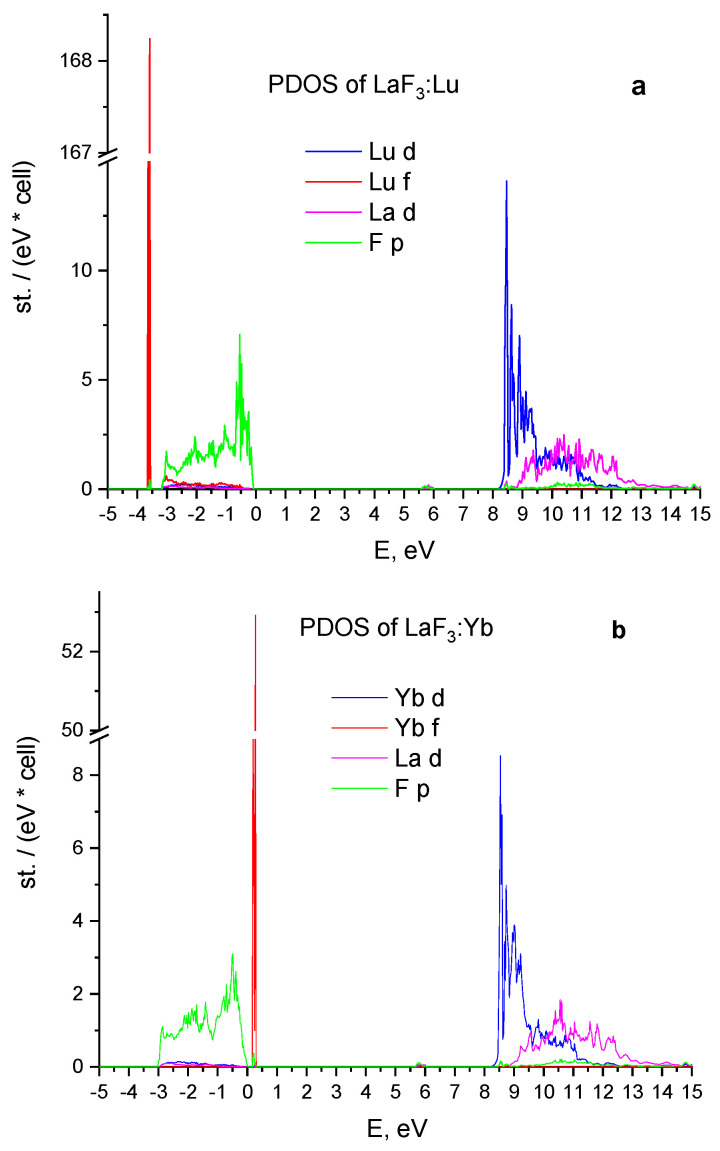
Partial density of states of the LaF_3_:Lu (**a**) and LaF_3_:Yb (**b**) crystals.

**Figure 3 materials-15-07937-f003:**
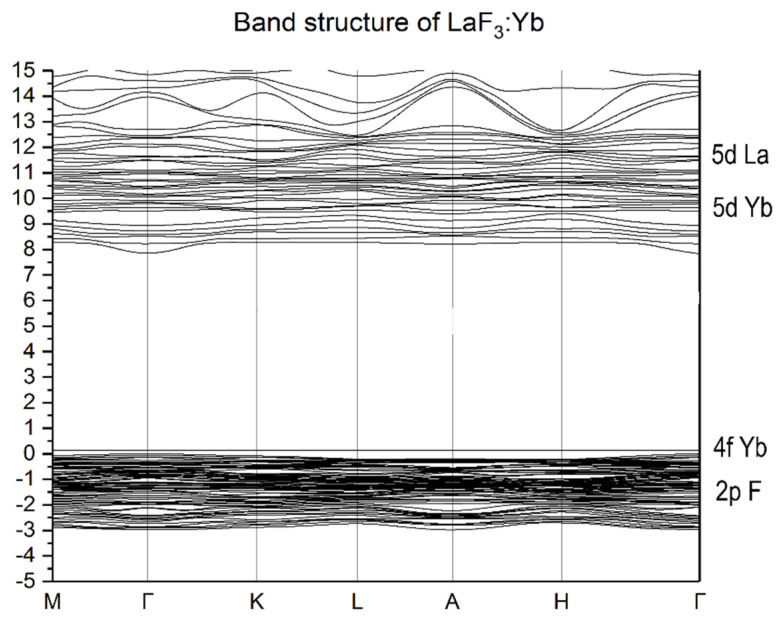
The energy band structure of LaF_3_:Yb.

**Figure 4 materials-15-07937-f004:**
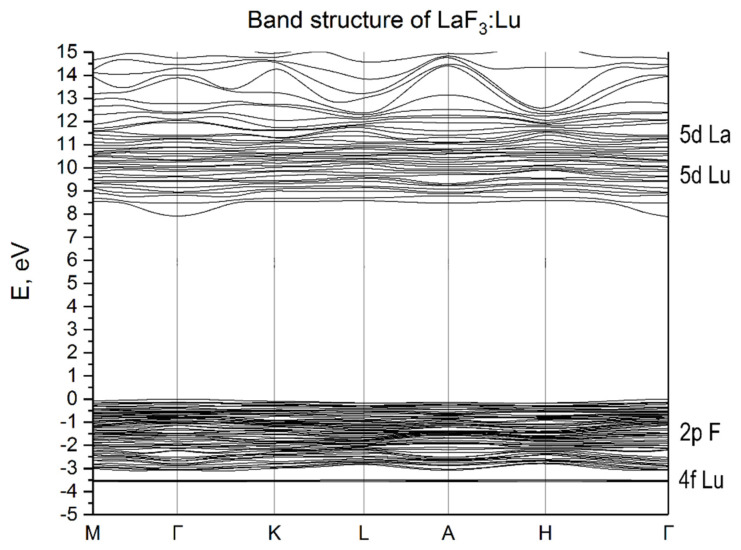
The energy band structure of LaF_3_:Lu.

**Figure 5 materials-15-07937-f005:**
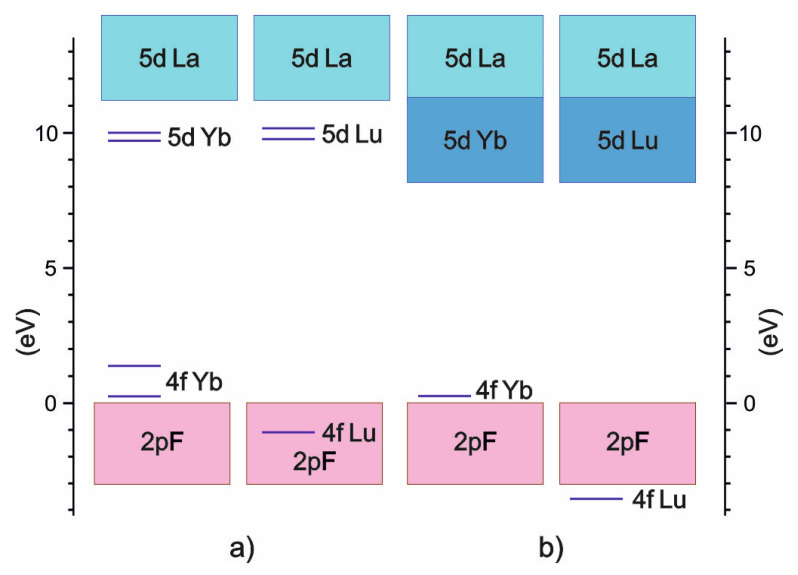
Scheme of energy levels of impurities in LaF_3_:Yb and the LaF_3_:Lu crystals ((**a**)—experimental results [2], (**b**)—calculated results).

**Table 1 materials-15-07937-t001:** Initial and optimized lattice parameters of the LaF_3_ crystal.

	a, Å	b, Å	c, Å	α	β	γ
Initial	7.25	7.25	7.39	90	90	120
Optimized	7.22	7.22	7.36	90	90	120

## Data Availability

The data presented in this study are available on request from the corresponding author (thanks xf@hotmail.com).

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
