# Peer review of "Computer Modelling of Energy Structure of Yb3+ and Lu3+ Doped LaF3 Crystals"

_materials, 2022, doi:10.3390/ma15227937_

Round 1
Reviewer 1 Report
In this manuscript, modelling of energy etructure of Yb3+ and Lu3+ doped LaF3 crystals was studied. The influence of geometric optimization on the results of energy band calculations of LaF3:Ln crystals (Ln = Yb, Lu) has been analyzed. The calculated band gap of 9.6 eV for LaF3 is in a good agreement with experiment result. The conclusions can be supported by data, however, the manuscript is of low quality. The reviewer think it can not be considered for publication for now.
1. The originality and importance was absent in the manuscript, or they are not detectable.
2. What kind of problems could be solved by this study? This was not described clearly in INTRODUCTION.
3. The language is terrible. There are many mistakes in expression. Please check through the manuscript.
Author Response
Dear Reviewer 1,
Thank you for reviewing the manuscript and valuable comments which will enhance the manuscript quality. Below you will find our answers.
Q1. 1. The originality and importance was absent in the manuscript, or they are not detectable
A1. Thanks for your comments. We revised the content of the introduction section and extended the literature review.
Q2. What kind of problems could be solved by this study? This was not described clearly in INTRODUCTION.
A2. Thanks for your comments. We took into account your recommendation and considerably revised the introduction section
Q3. The language is terrible. There are many mistakes in expression. Please check through the manuscript.
A3. Thanks for your comments. We asked a professional linguist to improve the linguistic aspect of our paper.
Reviewer 2 Report
The manuscript entitled “Computer Modelling of Energy Structure of Yb3+ and Lu3+ Doped LaF3 Crystals” by Yaroslav Chornodolskyy et al. fits in the Materials journal’s scopes. In the current state, this work can be accepted for publication.
The authors clearly identify the state of the art of the lanthanide ions based inorganic scintillators and their applications, tackling it with their contribution with a comprehensive theoretical analysis of the LaF3: Yb and LaF3: Lu crystals, more specifically theoretical calculations of the 4f and 5d level positions in LaF3 host matrix and the necessity to choose suitable theoretical strategies for energy level calculation of the lanthanide ions.
Author Response
Dear Reviewer 2,
Thank you for reviewing the manuscript and positive response.
Reviewer 3 Report
In the current article, authors reported the quantum chemical calculations of the energy band structure, partial and total densities of states for LaF3:Yb and LaF3:Lu crystals at the DFT+U level of theory. The work is interesting for the readers of Materials and the obtained results are important for the Materials Science. However, the article has serious issues that must be solved to make this article publishable:
1) Authors should compare the obtained results with the undoped LaF3, for example 10.1088/1361-648X/aac452
2) Section 3. Materials and Methods: authors should provide the basis set that was used in calculations
3) Lines 120-124: Authors should add the citation of the LaF3 structure that was used as guess structure for the LaF3:Yb and LaF3:Lu geometry optimization.
4) Lines 121-122: Authors should add the citation of the online resource “Materials Project”.
5) Authors should provide the optimized structure of the LaF3:Yb and LaF3:Lu crystals
6) Authors should mention the concentration of the Lu and Yb ions used in the calculations.
7) The ionic radius La3+ is significantly larger than that of Lu3+ and Yb3+, and therefore, partial replacement of La3+ to Lu3+ or Yb3+ should cause the structural defects and the symmetry reduction. Please discuss this effect in details.
8) Lines 189-190: Authors mentioned that “The calculated band gap in both the cases is 9.6 eV, which agrees well with the experimental data [33] and with the calculation results obtained for LaF3:Pm and LaF3:Sm [42].” Please, provide the band gap values for the experimental data and LaF3:Pm and LaF3:Sm.
Author Response
Dear Reviewer 3,
Thank you for reviewing the manuscript and valuable comments which will enhance the manuscript quality. Below you will find our answers.
Q1.Authors should compare the obtained results with the undoped LaF3, for example 10.1088/1361-648X/aac452.
A1. Thanks for your comments. We added the chapter 4.2 entitled “Energy properties of LaF3:Lu and LaF3:Yb ”. There is a citation of computation value for pure LaF3 .
Q2. Section 3. Materials and Methods: authors should provide the basis set that was used in calculations
A2. Thanks for your comments. In chapter «Materials and Methods» we added the parameters, which we use for computation of the zonal structure.
Q3. Lines 120-124: Authors should add the citation of the LaF3 structure that was used as guess structure for the LaF3:Yb and LaF3:Lu geometry optimization.
A3. Thanks for your comments. There is a link for free software product “Materials Project” for forming and optimizing the structures of LaF3:Yb and LaF3:Lu is given in the second paragraph of the chapter «Materials and Methods».
Q4. Lines 121-122: Authors should add the citation of the online resource “Materials Project”.
A4. Thanks for your comments. There is a link for free software product “Materials Project” for forming and optimizing the structures of LaF3:Yb and LaF3:Lu is given in the second paragraph of the chapter «Materials and Methods».
Q5. Authors should provide the optimized structure of the LaF3:Yb and LaF3:Lu crystals
A5. Thanks for your comments. In the third paragraph of chapter «Materials and Methods» and in the first paragraph of chapter “Results” we state that during the optimization only the positions of ions were changed but not the parameters of the lattice.
Q6. Authors should mention the concentration of the Lu and Yb ions used in the calculations.
A6. Thanks for your comments. In chapter «Materials and Methods» we added this piece of text in the second paragraph: “Due to the limited computer resource for calculation of impurity energy levels, we chose a model for which one of the lanthanide ions of the matrix unit cell was replaced by the ytterbium or lutetium ion, respectively.”
Q7. The ionic radius La3+ is significantly larger than that of Lu3+ and Yb3+, and therefore, partial replacement of La3+ to Lu3+ or Yb3+ should cause the structural defects and the symmetry reduction. Please discuss this effect in details.
A7. Thanks for your comments. In chapter «Materials and Methods» we added this piece of text in the second paragraph: ”To consider the difference between the ionic radius of lanthanum and ytterbium/lutetium, the procedure of geometry relaxation has been performed. New ions' positions have been calculated with the same lattice parameters as in the original matrix”.
Q8. Lines 189-190: Authors mentioned that “The calculated band gap in both the cases is 9.6 eV, which agrees well with the experimental data [33] and with the calculation results obtained for LaF3:Pm and LaF3:Sm [42].” Please, provide the band gap values for the experimental data and LaF3:Pm and LaF3:Sm.
A8. Thanks for your comments. The experimental value of the band gap for LaF3:Pm Ñ– LaF3:Sm is considered in the paper Dorenbos, P. Ce3+ 5d-centroid shift and vacuum referred 4f-electron binding energies of all lanthanide impurities in 150 different compounds. J. Lumin. 2013, 135, 93-104. doi: 10.1016/j.jlumin.2012.09.034. It is equal to 9,5 eV.
Reviewer 4 Report
Referee report on manuscript “Computer Modelling of Energy Structure of Yb3+ and Lu3+ Doped LaF3 Crystals” by Yaroslav Chornodolskyyet al
This version does not look worthy and cannot be recommended for publication in this form and at least needs very proper improvement and clarification.
1. The introduction is poorly written, and there is a clear feeling that it is written for another article. The first [1-31] references do not belong to the topic of the article. There is no real connection between the abstract and the introduction. And the introduction in its current form should be removed.
2. Paragraph "2. State of the art" - this can be an introduction, because it reveals why this work is being done. However, since this work is from theoretical modeling, recent similar work should be familiar to the authors and mentioned.
a) Chuklina, N., Piskunov, S., Popov, N. V., Mysovsky, A., & Popov, A. I. (2020). Comparative quantum chemistry study of the F-center in lanthanum trifluoride. Nuclear Instruments and Methods in Physics Research Section B: Beam Interactions with Materials and Atoms, 474, 57-62.
b) Valerio, M. E. G., Jackson, R. A., & De Lima, J. F. (2000). Derivation of potentials for the rare-earth fluorides, and the calculation of lattice and intrinsic defect properties. Journal of Physics: Condensed Matter, 12(35), 7727.
3. Band gap data needs to be compared with other calculations. See Table 2 in above-mentioned paper a). Therefore, comparative analysis with other calculation approaches in important.
Author Response
Dear Reviewer 4,
Thank you for reviewing the manuscript and valuable comments which will enhance the manuscript quality. Below you will find our answers.
Q1. The introduction is poorly written, and there is a clear feeling that it is written for another article. The first [1-31] references do not belong to the topic of the article. There is no real connection between the abstract and the introduction. And the introduction in its current form should be removed.
A1. We agree with your recommendation. We considerably revised and shortened this section. We added so extensive consideration of the interconnection between the applied and fundamental science because in Ukraine current policy in the field of science lies in the direction of separation of these two branches of science. We deeply disagree with this policy and decided to reflect the interconnection between the branches of science in the introduction section to show that science in in fact integral and the division into applied and fundamental science is rather conditional and made for convenience. We tried to show that collaboration between both branches of science enriches both of the and artificial division with a distinct border between them will only harm both branches of science. If they stay confined in their areas they will inevitably suffer due to shortage of ideas. Because current suggestions in the field of science in Ukraine are going to create clear classification of papers and scientists to be either fundamental or applied. And those who belong to fundamental science will have considerable difficulties with collaboration and participation in the projects from the field of applied science and vice versa. As a good example of successful career that of the scientists who worked on the borderline is the leader of this project professor Orest Kochan. He graduated with distinction from the faculty of physics at Ivan Franko Lviv National University but chosen as the field of interest the field of applied science. His successfully used his skills obtained during the training as a fundamental scientist to solve applied problems of temperature sensors but never abandoned his ties with the fundamental science. He managed to combine people from different background and experience and launch the project in the field of fundamental science within which we published one paper in the Materials last year (Kochan, O.; Chornodolskyy, Y.; Selech, J.; Karnaushenko, V.; Przystupa, К.; Kotlov, A.; Demkiv, T.; Vistovskyy, V.; Stryhanyuk, H.; Rodnyi, P.; Gektin, A.; Voloshinovskii, A. Energy Structure and Luminescence of CeF3 Crystals. Materials 2021, 14, 4243. https://doi.org/10.3390/ma14154243), prepared this particular paper and are on the verge of finishing a few more papers. If a new policy will be adopted, such projects of people with mixed areas will be very difficult if possible at all. That is why our previous version of the introduction was something like a political manifesto. But we are scientists and scientific papers are our production that is why we use them to promote our views. As real scientist we use our papers as our trumps against strange policies proposed by politicians. We can rely only upon scientific arguments that justify our position. If our papers with the text like we wrote in the introduction passes the review we use it as an argument that our views on the close interconnection between the fundamental and applied science was supported, or at least not confronted, by reviewers who are big experts in science. Thus the view of politicians contradict the view of scientists. That is why the opinion of the latter should be changes as scientists do not support the strong division between applied and fundamental science. Our credo can be summarized by the phrase “There is nothing so practical as a good theory”.
We revised our the introduction section but still we hope for your understanding and support. So we would kindly ask your permission to leave some elements of our manifesto in this paper as a part of introduction.
Q2. Paragraph "2. State of the art" - this can be an introduction, because it reveals why this work is being done. However, since this work is from theoretical modeling, recent similar work should be familiar to the authors and mentioned. a) Chuklina, N., Piskunov, S., Popov, N. V., Mysovsky, A., & Popov, A. I. (2020). Comparative quantum chemistry study of the F-center in lanthanum trifluoride. Nuclear Instruments and Methods in Physics Research Section B: Beam Interactions with Materials and Atoms, 474, 57-62.
- b) Valerio, M. E. G., Jackson, R. A., & De Lima, J. F. (2000). Derivation of potentials for the rare-earth fluorides, and the calculation of lattice and intrinsic defect properties. Journal of Physics: Condensed Matter, 12(35), 7727
A2. Thanks for your comments. We, according to your recommendation, combined the first and second chapters. We added the paper Valerio, M. E. G., Jackson, R. A., & De Lima, J. F. (2000). Derivation of potentials for the rare-earth fluorides, and the calculation of lattice and intrinsic defect properties. Journal of Physics: Condensed Matter, 12(35), 7727 to the list of references.
Q3. Band gap data needs to be compared with other calculations. See Table 2 in above-mentioned paper a). Therefore, comparative analysis with other calculation approaches in important.
A3. Thanks for your comments. We added in chapter 4.2 “Energy properties of LaF3:Lu and LaF3:Yb ”, at the fourth paragraph of it, the reference with the computational value for pure LaF3.
Round 2
Reviewer 1 Report
The authors have responded seriously to the original comments. There are no more suggestions for the manuscript.
Author Response
Dear Reviewer,
Thank you for your very important and valuable comments.
Best Regards,
Krzysztof Przystypa
Reviewer 3 Report
Authors revised the manuscript according to me suggestions and now it can be accepted for publication.
Author Response

(The authors gave the same response as above.)

Reviewer 4 Report
This manuscript is absolutely unsuitable for publication and needs very major revision.
1. The description of the calculations and calculated results is inadequate. First of all, quite often in the text a wrong terminology is used, which indicates a low degree of familiarity of the authors with the subject they are discussing. The density of states is not the "energy parameters", as the authors state in Part 3, but rather refer to the electronic properties of a studied compound. It is not clear what was the size of a supercell used for the calculations, and the authors' statement (line243) "we chose a model for which one of the lanthanide ions of the matrix unit cell was replaced by the ytterbium or lutetium ion" is too vague to shed some light on the adopted structural model. Please describe explicitly the size of the model, the concentration of the dopants. It also would be useful to evaluate lateral interaction of dopants.
Why did not you try full geometry optimization (atom positions + cell parameters)?
2. Another unclear statement (line 354): "New ions' positions were calculated with the same lattice parameters as in the original matrix" - were the ions allowed to relax? Were the lattice constants fixed (which would be wrong?) The right approach would be to allow to change BOTH lattice constants and ionic positions.
3. Table I caption is misleading – it just states "Table 1. Lattice parameters" – of what? Or which lattice parameters – experimental, optimized or whatever? By the way, were the calculations spin-polarized? Moreover, this table must necessarily contain all the literature data (both computational and experimental).
4. Another unclarified statement (line 446): "The 5d Yb3+ states partially overlap with the conduction band states and have a typical shape for the case of splitting by a crystal field" – did the authors estimate this crystal field splitting? Does it in reality follow the pattern prescribed by theory – I mean, the order of the split 5d states and the intervals between them?
5. Another pitfall: "The first is the presence of 4f state above the top of the valence band in LaF3:Yb" – Yb3+ ions have 4f^13 configuration, and there should be 14 (with spin) unoccupied 4f states. How many state are above the valence band?
6. One more inaccuracy: "t small scattering of energy 5d-bands located below the bottom" – perhaps, the authors meant a low dispersion of the impurity states? Another improper term.
Here and there are many examples of improper English use, e.g. "the conditions for local states such as Frenkel (line 472) excitons appear" – the conditions can be rather formed, but not appear. English should have been thoroughly polished through the whole paper. Fig. 5 – sometimes the authors give only one 4f level of Yb, sometimes – two. Why?
7. Line 508. The statement "The positions of the 5d levels in LaF3:Yb and the LaF3:Lu crystals were shown to be energetically close and placed below the conduction band" can be made if and only if both schemes are linked to the same reference point (vacuum level), otherwise, the closeness of those 5d states to the conduction bands of two different crystals says nothing.
"The small energy gap in the (line 509) proximity of the Г-point in case of the LaF3:Yb crystal was revealed" – what kind of a gap? Between what states – it is not specified.
The authors should understand that they try to analyze the properties of multielectron ions, using essentially one-electron based DFT calculations. This can be done only with utmost care and complete understanding of possible drawbacks.
8. The authors ignored the previous recommendation to make a detailed comparison of band gap Eg with other literature data. See Table 2 and the corresponding text in:
Chuklina, N., Piskunov, S., Popov, N. V., Mysovsky, A., & Popov, A. I. (2020). Comparative quantum chemistry study of the F-center in lanthanum trifluoride. Nuclear Instruments and Methods in Physics Research Section B: Beam Interactions with Materials and Atoms, 474, 57-62. And references therein.
For these additional Table is required.
9. One more additional Table is required, namely, showing the displacements of the nearest atoms near the impurity
10. In the revised version Section 2 is now missing.
11. "ASE software" Reference is missing.
12. Line 395. However, the position of 5d Lu3+ relative to the conduction band is better in the un-relaxed model." This does not mean that it is better not to optimize the structure, but that the model is not very precise.
13. "The calculated band gap created by the 5d states of La3+ is 9.6 eV for both crystals." This is an odd conclusion, the band gap should not be affected by a dopant in low concentration.
14. The first paragraph of the introduction is unclear and off-topic, references [1-4] are considered INCORRECT citations and should be removed.
Note that this manuscript can be recommended for publication ONLY after a detailed consideration and disclosure of the all above-mentioned ambiguities.

Author Response
The answer is attached in an additional file.
